# Network disruption via continuous batch removal: The case of Sicilian Mafia

**Mingshan Jia**[1]*, **Pasquale De Meo**[2], **Bogdan Gabrys**[1], **Katarzyna Musial**[1]

**1** School of Computer Science, University of Technology Sydney, Sydney, NSW, Australia, **2** Department of Ancient and Modern Civilizations, University of Messina, Messina, Italy

* mingshan.jia@uts.edu.au

## Abstract

Network disruption is pivotal in understanding the robustness and vulnerability of complex networks, which is instrumental in devising strategies for infrastructure protection, epidemic control, cybersecurity, and combating crime. In this paper, with a particular focus on disrupting criminal networks, we proposed to impose a within-the-largest-connected-component constraint in a continuous batch removal disruption process. Through a series of experiments on a recently released Sicilian Mafia network, we revealed that the constraint would enhance degree-based methods while weakening betweenness-based approaches. Moreover, based on the findings from the experiments using various disruption strategies, we propose a structurally-filtered greedy disruption strategy that integrates the effectiveness of greedy-like methods with the efficiency of structural-metric-based approaches. The proposed strategy significantly outperforms the longstanding state-of-the-art method of betweenness centrality while maintaining the same time complexity.

**Data Availability Statement:** https://www.kaggle.com/datasets/mingshanjia/sicilian-mafia-criminal-network.

**Funding:** This work was supported by the Australian Research Council, Grant No.

## 1 Introduction

The study of complex networks has emerged as a crucial interdisciplinary field, providing insights into systems as diverse as social interactions, biological processes, infrastructural designs, financial markets, transportation networks, and communication systems [1–3]. Within this realm, understanding network disruption strategies is of paramount importance, particularly for mitigating the resilience of undesirable networks such as criminal organisations or disease-spreading pathways [4, 5]. The task of disrupting a network can be formulated as the task of finding the smallest set of nodes that, if removed from the network, determine the largest decrease in the size of the largest connected component (LCC, for short) of the network. Unfortunately, previous studies show that such a problem is NP-hard [6–8].

Existing approaches to network disruption can generally be classified into three categories. The first category of approaches uses structural metrics, such as degree or betweenness centrality, to rank nodes and progressively delete them from the network according to their rank [9–11]. The second category of methods starts by identifying nodes on the basis of their specific role within the network, such as those nodes that lie between predefined communities [12], the so-called articulation points whose removal would disconnect the network [13] or the

DP190101087: Dynamics and Control of Complex Social Networks. the funders had no role in study design, data collection and analysis, decision to publish, or preparation of the manuscript.

**Competing interests:** The authors have declared that no competing interests exist.

nodes of high k-shell indices [14]. These selected nodes are then further ranked according to metrics like degree or betweenness centrality. The third set of approaches for disrupting a network formulates the disruption process as a two-step procedure: the initial step focuses on selecting nodes to decycle the network, followed by choosing nodes to break the resulting tree structures [6, 15]. Surprisingly, despite the efficacy of various newly developed techniques in specialised network contexts, betweenness centrality is still shown to be the most universally effective strategy for network disruption [16].

In this work, we further investigate the disruption strategies in the context of criminal networks [17]. Specifically, we focus on a Sicilian Mafia Network [18, 19]. Unlike conventional criminal organisations, Mafia clans are characterised by unique structural attributes: they are composed of loosely coupled groups that can span across multiple generations [20]. These groups, often referred to as "coscas", "families" or "clans", are bound by strong relational ties and reciprocal altruism. Mafia groups can have a profound influence over economic, social, and political sectors in some countries [21]. Therefore, finding a way to effectively disrupt mafia clans is crucial not only for academic research but also serves as a practical guide for Law Enforcement Agencies (LEAs) in devising targeted countermeasures.

Due to the inherent structural characteristics of criminal networks and the constraints of limited police resources, attacks against such networks are often not accomplished in one stroke. Mafia gang is strongly rooted in a specific social and professional environment, and such an environment acts as a protective shield over the gang leaders (also known as *bosses*) and its members. A boss lives in disguise and often enlists the help of people who are not affiliated with the gang (e.g. doctors and nurses who provide medical treatments to the boss without revealing her/his identity to the police). Efforts to catch mafia bosses therefore take many years and they start with people who support the activities of a mafia gang without being part of the gang; police forces try to catch the boss by investigating and arresting people who are "closer and closer" to the boss. Police proceeding ends with the arrest of the boss along with a circle of accomplices and collaborators and, consequently, a peculiarity of the process of dismantling a criminal network is that the network is not attacked one node at a time, but the investigations simultaneously target groups of nodes. Therefore, we view criminal network disruption as a continuous batch removal process [18]. In this framework, a specified number of target nodes are removed at each iterative step. This approach extends traditional strategies, which typically assume the removal of only a single node at a time. Also, given the constraints of limited police resources, it may be necessary to concentrate efforts in a specific location. As such, we also examine the impact of various disruption strategies constrained to the LCC, rather than targeting the whole network. We evaluated seven distinct disruption strategies within the context of continuous batch removal on the mafia network: degree centrality [9], betweenness centrality [22], collective influence [11], CoreHD [15], APTA [13], as well as variations of degree and betweenness centrality that incorporate secondary ranking criteria for instances of tied rankings. We discover that betweenness centrality and APTA emerge as the most effective strategies when there are no constraints limited to the LCC. On the other hand, degree centrality performs exceptionally well within the LCC constraint, especially when the batch size is small.

In pursuit of a strategy that could surpass traditional approaches, we initially adopted a greedy algorithm. In each iteration, our algorithm identifies and removes the batch of nodes resulting in the greatest reduction of the LCC size. As anticipated, the greedy strategy significantly outperforms classical methods, particularly when the batch size exceeds one. However, the greedy approach is computationally infeasible for large networks or large batch sizes. To address this issue, we optimise the greedy algorithm by narrowing the search space. Our study suggests that a large proportion of nodes targeted by the greedy method also rank high in both

betweenness and degree centrality, and selected nodes are often articulation points. Thus, we introduce a novel approach, called Structurally-Filtered Greedy Approach (SF-GRD), which restricts the search space to nodes that are highly ranked in these metrics. The results show that SF-GRD achieves a disruption effect comparable to the greedy algorithm while maintaining the same time complexity as betweenness centrality.

To summarise, our contribution in this work is manifold:

- We propose a generalised robustness metric that is more suitable for effective disruption strategies in a continuous batch removal setting;

- Focusing on the disruption of the mafia network and aiming to mimic actual police raid operations, we propose to apply the constraint of within-LCC in a continuous batch removal setting;

- We elucidate the disparate impacts of the within-LCC constraint on degree centrality and betweenness centrality;

- After analysing the result of a naive greedy approach, we propose a Structurally-Filtered Greedy approach that not only outperforms traditional disruption strategies but also maintains the same polynomial time complexity as betweenness centrality.

The remainder of this paper is organised as follows: Section 2 summarises current approaches in dismantling criminal networks; Section 3 explores and proposes metrics for assessing network disruption processes; Section 4 presents our case study dataset, along with requisite preprocessing steps and initial insights; Section 5 outlines classical disruption strategies, introduces the within-LCC constraint, and provides a comprehensive discussion of the results; Section 6 describes the Structurally-Filtered Greedy approach (SF-GRD), demonstrating its effectiveness and efficiency; and finally, Section 7 offers concluding remarks.

## 2 Related literature

Criminal and covert networks are flexible organisations that can quickly adapt their organisation and behaviour to survive. However, empirical research on the dynamics of criminal networks is still limited [18, 19, 23, 24] due to the lack of accurate and complete datasets. Simulation studies have shown that removing actors in the most central positions in the criminal network is one of the most efficient strategies [23].

The main limitation of existing node removal processes stems from the fact that these processes simulate the arrest of one individual at a time. In contrast, law enforcement often arrests both the leader of the organisation (referred to as the *boss*) along with several accomplices who helped the boss escape (e.g., by protecting her/his anonymity).

We also point out that attempts to remove one or more nodes from a criminal network generally do not lead to the dismantling of the organisation, but only its weakening. Gangs are able to reorganise quickly by, for instance, electing new leaders, changing the communication procedures as well as the rules for recruiting new members. Paradoxically, a criminal organisation can become stronger and more cohesive after its reorganisation [25–27].

In this paper, we focus on the size of the largest connected component (LCC) of the graph(s) associated with a mafia gang: indeed, an LCC that covers almost all the nodes of a criminal gang allows all its members to communicate and coordinate to carry out illicit tasks; a significant reduction in the LCC severely inhibits the ability of gang members to communicate and, consequently, severely damages the criminal power of the gang. The activities required to rebuild the LCC are costly and dangerous because, for example, the gang must elect new members with the aim of connecting separated sub-communities. Such an activity

is hard to implement in the short term because the potential mediators need to be trusted individuals from all the involved sub-communities. Furthermore, the police have to deal with smaller sub-communities, it is easier to thwart gang reconstitution attempts. In this study, we propose strategies to reduce the size of the LCC and, from a technical point of view, our problem is similar to some well-known problems in graph theory concerning the decomposition of graphs into connected components. More specifically, a *vertex cut* of a graph $G = \langle V, E \rangle$ is a subset of nodes $V'$ of $V$ such that the graph $G'$ obtained by deleting nodes in $V'$ from $G$ has at least $k$ (non-empty and pairwise disconnected) components [8, 28, 29]. The *vertex k-cut problem* requires constructing a vertex $k$-cut of minimum cardinality, if it exists, and, as shown by Cornaz *et al.* [8], such a problem is NP-hard for any $k \geq 3$. The $k$-cut problem consists in finding the smallest set of edges $E'$ to remove from $G$ to produce at least $k$ connected components [30, 31]. Cornaz *et al.* [8] solved the $k$-VC problem using a general-purpose integer linear programming solver; more recently, Zhou *et al.* developed a fast local search algorithm to solve it [32].

Our work advances the state of the art by proposing an algorithm to find the nodes whose removal gives the largest reduction in the LCC. Our approach is much more accurate than current best betweenness-based methods and maintains the same computational complexity. Since criminal networks extracted from court documents typically contain around a hundred nodes, the betweenness computation can be performed efficiently, making our approach suitable for supporting law enforcement investigations.

## 3 Measuring disruption processes in networks

In the analysis of network behaviour under different disruption scenarios, particularly when evaluating robustness or vulnerability, it is crucial to employ measures that encapsulate the specifics and impact of the disruptions. This is essential across a range of fields, such as controlling the spread of diseases, investigating the propagation of information in social networks and understanding the structural integrity of criminal networks. Two main measures have been proposed to quantify the effectiveness of disruptions, each with its own merits and applicability.

### 3.1 One-time disruption

The first category of disruption measure is proposed to evaluate a one-time disruption [6, 9]. The objective is to minimise the number of nodes that need to be removed to reduce the size of the LCC below a certain threshold $N^* = qN$, where $N$ is the number of nodes in the network, and $q$ is a fraction between 0 and 1. More formally, the optimisation problem is outlined below:

$$\min_{S \subseteq V} |S| \quad \text{subject to} \quad |LCC(V \setminus S)| \leq N^*, \tag{1}$$

where $S$ is the set of node targeted by a given strategy. This way, the size of $S$ is used to evaluate the effectiveness of a one-time disruption: the smaller the size, the more effective the strategy is.

### 3.2 Continuous disruption

The second category is the *robustness measure* and they are proposed to evaluate a continuous disruption process where nodes are progressively removed from the network [16, 33].

The *robustness R* is defined as the average of the normalised sizes of the LCC after each node's removal, until all nodes are removed, which is:

$$R = \frac{1}{N} \sum_{i=1}^{N} \frac{L(i)}{N}, \tag{2}$$

where $N$ is the total number of nodes, and $L(i)$ is the size of the LCC after removing $i$ nodes. The robustness $R$ is bounded at $[\frac{N-1}{N^2}, \frac{N-1}{2N}]$. The lower bound is reached in a star graph and when the first removed node is the center of the star; while the upper bound is taken at a complete graph (LCC is viewed as zero when all nodes are removed in the calculation).

The robustness measure not only provides a means to evaluate the inherent resilience of a network structure to node removal, but also serves as a quantitative tool to compare the effectiveness of different disruption strategies. It captures the progressive nature of the disruption process, reflecting how well the network can maintain its connectivity as nodes are sequentially removed. By analysing changes in $L(i)$ in different strategies, we can identify which approaches more effectively degrade the network's robustness.

## 3.3 Robustness of partial removal

Although the original robustness measure tracks the entire disruption, it is very likely we care more about the effectiveness of only removing a subset of network nodes. Many effective disruption strategies would rank the nodes according to their structural attributes, and the removal of a small percentage of top-ranked nodes would significantly break the network into very small components, making the removal of the remaining nodes insignificant. For example, when applying a betweenness-degree disruption strategy on a real-world criminal network [18], the network has been significantly broken after 20% of nodes are removed—LCC size has dropped to 7, about just five percent of the original LCC size; and the network becomes completely disrupted when 40% of nodes are eliminated.

Therefore, we propose a novel robustness measure for partial removal, denoted $R@r$, which aims to capture the effects of removing a small percentage of nodes in the network. Furthermore, we also generalise the removal process to support batch removal, where at each step, more than one node can be removed. Batch removal process is also a better reflection of the police raid operation in the real world, as combating organised crime often requires continuous actions, and the effectiveness (the number of nodes removed) that can be achieved by each operation is related to the deployable police force.

Specifically, with a batch size $b$, the robustness measure $R@r$ is defined as the average of the normalised sizes of the LCC for each removal step, up to $r$ percent of total nodes removed. Mathematically, it is given by:

$$R@r = \frac{1}{s} \sum_{i=1}^{s} \frac{L(b*i)}{N}, \qquad s = \frac{r*N}{b}, \tag{3}$$

where $N$ is the total number of nodes, $s$ is the number of steps until a fraction $r$ of total nodes are removed, and $L(b*i)$ is the size of the LCC after $b*i$ nodes are removed after step $i$. Clearly, when the percentage $r$ equals 1, and the batch size $b$ equals 1, it degrades to the original robustness measure $R$.

This measure provides a focused view of the initial stages of the disruption process, which are often the most critical, allowing us to better evaluate the performance of different disruption strategies.

## 4 Dataset description

In this section, we introduce our case-study dataset, the preprocessing steps we performed as well as some basic characteristics of the dataset. Our dataset is available at www.kaggle.com/datasets/mingshanjia/sicilian-mafia-criminal-network.

The dataset was derived from the pre-trial detention order issued by the Court of Messina's preliminary investigation judge on March 14, 2007, which was towards the end of the major anti-mafia operation referred to as the "Montagna Operation". The particular investigation was a prominent operation focused on two mafia clans, i.e., the "Mistretta" family and the "Batanesi" clan. From 2003 to 2007, these families were found to have infiltrated several economic activities, including major infrastructure works, through a cartel of entrepreneurs close to the Sicilian Mafia.

The pre-trial detention order reported the composition of both the Mistretta family and the Batanesi clan; judicial proceedings followed a chronological narrative scheme which detailed the illicit activities pursued by the members of the Mistretta clan before the imprisonment of a boss (denoted as Boss X to preserve anonymity). The Boss X has been selected by the most influential mafia syndicates to settle the conflicts between the Mistretta and the Batanesi families. The conflicts were unleashed from the extortion imposed on local entrepreneurs in the construction of the highway that connects the city of Messina to Palermo.

Two graphs were originally constructed from the court order [34]: A meeting network and a phone call network (original networks are available at https://doi.org/10.5281/zenodo.3938818).

The meeting network represents physical encounters, while the phone calls network depicts telecommunication interactions. Meetings involving suspected individuals can be broadly classified as follows: (a) meetings that aim to define the organisational structure of the mafia syndicate along with its relationships with entrepreneurs and other mafia syndicates operating in surrounding areas. Boss X always attended these meetings and has always been accompanied by at least two trusted men. (b) Meetings involving people who occupied the lowest levels of the mafia syndicate hierarchy; the purpose of these meetings was, in general, to design illicit activities and usually, only two people were involved in these meetings. For each one, the date of the meeting, the place, and the participants were recorded. Participants in a meeting were identified by a unique code. The procedure to build the Meetings network was as follows: (a) Each person who participated in at least one meeting corresponds to a node in the network; (b) two subjects in the Meetings network are connected by an edge if both of them attended at least one meeting; (c) edge weights reflect the number of meetings that two individuals jointly attended. The phone call network is built in a similar manner.

Although the meeting and phone-call graphs individually offer valuable insights into the social dynamics among the subjects under investigation [19], they each capture only a subset of the interactions. Remarkably, these two graphs share about 50% of their nodes, indicating that they are not isolated spheres of activity but rather interlinked facets of a more complex organisation. To harness the full range of interaction types, we propose to unify these separate but overlapping networks into a single, comprehensive network: distinct edges from the two networks are added to the new network, and when edges appear in both the meeting and phone-call networks, their weights are summed up as the new weight for the edge in the integrated network.

This consolidated graph, which we name the *Unified Graph*, allows for a more accurate and comprehensive understanding of the underlying criminal organisation, covering both formal and casual interactions among individuals. More importantly, the Unified Graph can serve as a more effective platform for applying network disruption strategies. By targeting key nodes in

**Table 1. Network description and comparison.** The table lists six network metrics for seven different networks. The original disconnected graphs for meetings and phone calls are labelled as Meeting-DC (1st column) and Phone-Call-DC (2nd column), respectively. Their union forms the Unified-DC network (3rd column). After isolating the largest connected components, we obtain the Meeting Graph (4th column), Phone-Call Graph (5th column), and Unified Graph (6th column). The final column presents averaged statistics for 1,000 random graphs, each having the same number of nodes and edges as the Unified Graph.

| Network Metrics | Meeting-DC | Phone-Call-DC | Unified-DC | Meeting | Phone-Call | Unified | Random |
|---|---|---|---|---|---|---|---|
| Number of Nodes | 95 | 94 | 143 | 86 | 85 | **134** | 134 |
| Number of Edges | 248 | 120 | 326 | 242 | 113 | **320** | 320 |
| Avg. Degree | 5.22 | 2.55 | 4.56 | 5.63 | 2.66 | **4.78** | 4.78 |
| Diameter | - | - | - | 6 | 7 | **6** | 6.64 |
| Avg. Path Length | - | - | - | 3.11 | 3.33 | **3.11** | 3.29 |
| Avg. Clustering Coef. | 0.67 | 0.12 | 0.46 | 0.70 | 0.10 | **0.46** | 0.03 |

this unified network, law enforcement agencies can potentially dismantle or destabilise multiple facets of the criminal enterprise simultaneously, thereby enhancing the efficacy of their interventions.

After carefully examining the dataset, we introduced three data-preprocessing steps to address some data issues in the original edge lists:

- Retaining the larger weights. We detected inconsistent edge weights in the original edge list data, 29 cases in the meeting list, and 22 cases in the phone-call list, respectively. To deal with weight inconsistency, we chose to keep the larger value that represents the strength of the connection between two individuals.

- Keeping only the largest connected components. All three networks contain several tiny cliques and individual nodes that are disconnected from the LCC. We chose to exclude them from each network and focus on the core component of the criminal organisation.

- Removing self-loops. We removed self-loops from the data as they are not meaningful in the context.

After performing the preprocessing steps above, we obtained three key criminal networks: the Meeting Graph, the Phone-Call Graph, and the Unified Graph. Their features are summarised in Table 1. Initially, the Meeting-DC and Phone-Call-DC graphs consist of 95 and 94 nodes, which decrease to 86 and 85 nodes, respectively, in their connected versions. This suggests that the connected versions are comprised of the major components of the original graphs, capturing the core interactions, which is also evidenced by the marginal change in edge count.

Moving on to the specific graphs, the Meeting Graph has a higher average degree (5.63) and clustering coefficient (0.70) compared to the Phone-Call Graph, which has values of 2.66 and 0.10, respectively. This suggests that meetings involve more densely connected individuals, whereas phone calls occur in a more dispersed manner. After effectively merging them into the Unified Graph, the average degree (4.78) and clustering coefficient (0.46) fall between those of the Meeting and Phone-Call graphs.

The visualisation of the Unified Graph is given in Fig 1. As human capital is another important dimension to consider in criminal networks [35], we take advantage of the accompanying node feature information and categorise individuals based on both their familial and leadership roles. The details of this classification are in Table 2.

Lastly, when we compare the Unified Graph to randomly generated graphs with the same number of nodes and average degree, we observe significant differences. The Unified Graph has a much higher clustering coefficient (0.46 compared to 0.03) and also a smaller average

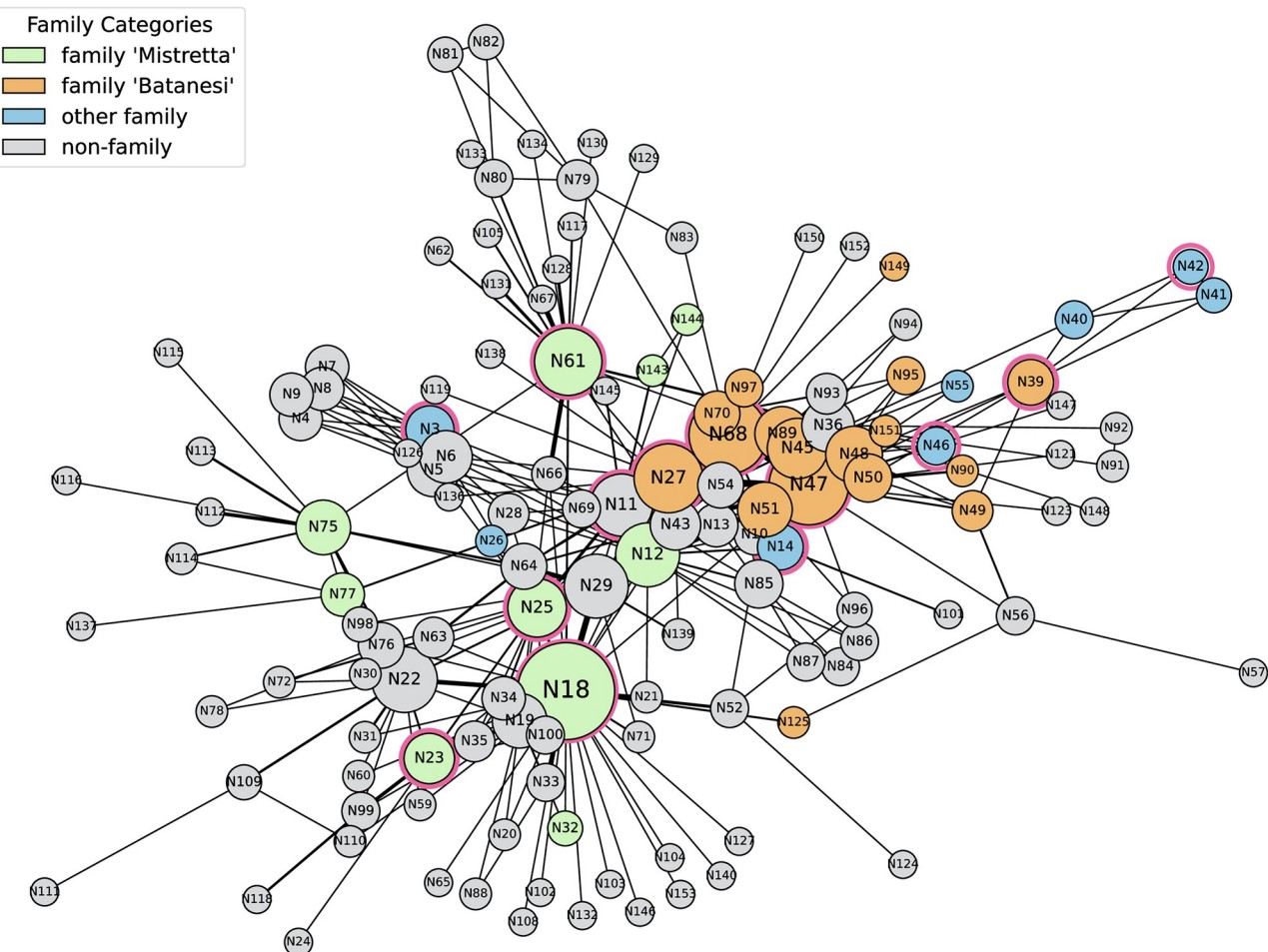

**Fig 1. The Unified Graph.** Different colours represent different clans within the mafia network. A red circle around a node signifies leadership roles within the network, including positions such as bosses and executives. Additionally, the size of a node is proportional to its degree, and the width of a link is proportional to its weight.

path length (3.11 compared to 3.29). These features indicate a tightly-knit community structure with efficient pathways for information or influence to flow. These are key considerations for understanding the complexities inherent in criminal activities and for designing effective intervention strategies.

**Table 2. Classification of nodes based on family and leadership roles.**

| Family Role | Count |
|---|---|
| Batanesi Family | 17 |
| Mistretta Family | 10 |
| Other Family | 8 |
| Non-family | 99 |
| **Leadership Role** | **Count** |
| Leaders | 13 |
| Non-leaders | 121 |

## 5 Disruption strategies

In this section, we describe strategies for efficiently disrupting a network. We recall that the problem of finding the smallest subset $S \subseteq V$ of a network $G = \langle V, E \rangle$ that, when removed from $G$, yields a graph $G'$ which largest connected component has size at most a fixed threshold $C$ is NP-complete [6]. A brute force solution would consider every possible subset of the nodes, resulting in a factorial time complexity.

To efficiently tackle the network disruption problem, we need to design approximate strategies that select the most promising nodes according to some metric (typically a measure of centrality) and progressively remove those nodes from the network.

In the following, we first introduce seven mainstream disruption strategies and then discuss how we modify and apply them to our study.

### 5.1 Disruption strategies

The first three attack strategies are derived from well-studied centrality metrics and they rank nodes according to a pre-defined metric and progressively remove nodes according to the ranking. CoreHD, APTA and MBA consider specific sub-structures within a graph (e.g. the 2-cores, the connected components, or the communities of that graph) to determine which nodes to remove. GND, on the other hand, utilises a spectral-cut method focusing on the graph's spectral properties.

- *Degree centrality (DEG)* [9]. This attack simply ranks nodes according to their importance measured by the degree centrality. The degree centrality of a given node $i$ is calculated by $C_D(i) = \sum_{j \in V} a_{ij}$, where $A = (a_{ij})$ is the adjacency matrix. We can consider two variants of the DEG strategy: the first is to compute degree centrality only once on the original graph; the second is to recompute degree centrality after each removal, thus accounting for dynamic changes in the perturbation process. We choose the recalculated degree approach to maximise its effectiveness in a continuous batch removal setting. That is, a number of nodes with the highest degrees are removed at each step until a predetermined total percentage of nodes are removed.

- *Betweenness centrality (BTW)* [22]. Betweenness centrality measures nodes' importance by measuring the number of times a node appears in the shortest path between all pairs of nodes and, thus, it can be regarded as a global centrality metric as it requires the full knowledge of the graph topology. The betweenness centrality of a node $i$ is defined as:

$$C_B(i) = \sum_{s,t \in V} \frac{\sigma(s, t \mid i)}{\sigma(s, t)}, \tag{4}$$

where $\sigma(s, t \mid i)$ is the number of shortest paths between $s$ and $t$ passing through node $i$, and $\sigma(s, t)$ is the number of all shortest paths between $s$ and $t$. As for the DEG strategy, we adopt the recalculated betweenness centrality to remove the top-ranked nodes at each step.

- *Collective Influence (CI)* [11]. Collective influence is a recently proposed centrality metric which takes into account the degree of nodes within a given radius when evaluating a node's impact on network disruption; intuitively, a node close to more high-degree nodes will have a higher collective influence score. Specifically, the collective influence $CI(i)$ of a node $i$ is computed as follows:

$$CI_\ell(i) = (k_i - 1) \sum_{j \in \delta B(i, \ell)} (k_j - 1) \tag{5}$$

where $k_i$ is the degree of node $i$, $B(i, \ell)$ is the ball of radius $\ell$ centered on node $i$. We define $B(i, \ell)$ as the set of nodes $x$ at a distance at most $\ell$ from $i$. Here the distance of two nodes is defined as the length of the shortest path joining them, and $\delta B(i, \ell)$ is the *frontier* of the ball, that is, the set of nodes at a distance $\ell$ from $i$. Essentially, $j$ represents nodes that are $l$-hop neighbours of $i$. In our experiment, $l$ is set to 1, and the ranking is recalculated after each removal.

- *CoreHD* [15]. By prioritising the decycling of a network, CoreHD focuses on the 2-core [36] when selecting a node to remove—more specifically, remove the highest-degree node from the 2-core at each step and, if more than one node has the same highest degree, one of them is randomly chosen. When 2-core no longer exists, the highest-degree node in the remaining tree-like graph is removed. In our batch-removal setting, CoreHD is extended to support multiple node removals at each step.

- *Articulation Point Targeted Attack (APTA)* [13]. An Articulation Point (AP) is any node in a graph whose removal increases the number of connected components. The removal of an AP node will split a connected component into multiple smaller connected components. APs can be identified in linear time using Tarjan's Algorithm [37]. APTA is a greedy AP attack that removes the most destructive AP (that is, the one causing the largest decrease in LCC size) at each step. APTA is similar to CoreHD in that they both limit the pool of targeted nodes. However, unlike from the above procedures, the "ranking" of AP nodes is no longer given by a structural metric, but is determined with the goal of minimising the size of LCC. We further modify the APTA attack so it can support multiple AP node removal. In cases where APs don't exist, nodes of the highest degree or nodes identified by another designated structural metric will be removed instead.

- *Module-based Attack (MBA)* [12]. MBA proposes to first identify topological communities within the network using a well established heuristic algorithm for community detection (Louvain algorithm is used in the experiment [38]). The nodes involved in links between different communities are then removed, starting with the one that has the highest betweenness centrality and proceeding in descending order. In order to make it suitable for batch removal, i.e., to dealing with the situation where the batch size exceeds the number of nodes with inter-community links, we choose to then target nodes based on their degree.

- *Generalised Network Dismantling (GND)* [7]. GND is a spectral-cut based strategy, where the network is dismantled by selectively targeting and removing certain nodes based on their spectral properties. GND operates by leveraging the spectrum of the graph Laplacian, particularly focusing on the Fiedler vector, i.e., the eigenvector associated with the second-smallest eigenvalue of the Laplacian matrix. This vector provides insights into the weak links of a network that, if removed, would lead to the network's fragmentation into disconnected components. GND is shown to be efficient for network dismantling. However, the disruption process is stochastic due to the random nature of spectral operation. Moreover, GND can not be used when network becomes very sparse because there is no meaningful Fiedler vector to work with. We also extend GND for batch removal in the following experiment.

## 5.2 Equality and localisation in disruption strategies

Building upon the five popular disruption strategies outlined above, there are still opportunities to enhance the efficiency of these approaches. Two specific enhancements can be considered to improve these strategies, namely:

1. *Introduction of a secondary ranking criterion.*
   Our first observation is that nodes could have the same centrality value, and in such a case, we would not be able to identify the next target node. An effective strategy is to combine a secondary criterion for node removal with our primary criterion: for example, we could select nodes based on their betweenness, and if two or more nodes have the same betweenness, we would favour nodes with the highest degree. The resulting strategy is called *BTW-DEG* (see Fig 2 for more details). The use of a secondary criterion is also useful in non-centrality-based node removal approaches (i.e. CoreHD and APTA): if Core or AP do not exist, nodes can still be selected according to the secondary criterion.

2. *Selection within the LCC.* Second, by narrowing the selection of nodes to those within the LCC rather than targeting nodes across the entire graph, we can create a more focused and efficient disruption process. Since the objective is to achieve the minimum robustness score (Eq 3), focusing on dismantling the LCC could enhance the effectiveness of the disruption, especially when only one node is removed at each step. Furthermore, this approach resonates with real-world scenarios, such as a police raid operation targeting a specific location, making it a practical and meaningful strategy for network disruption.

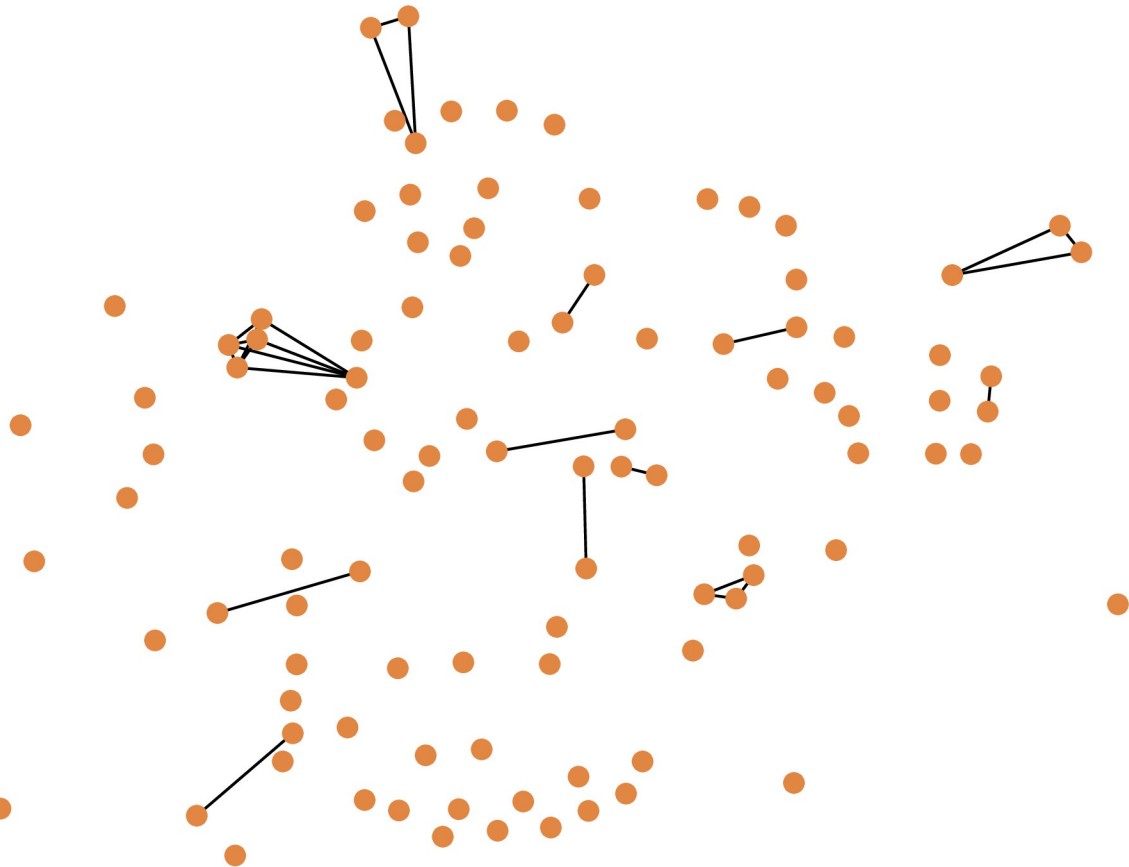

**Fig 2. A snapshot of a graph during the BTW-DEG disruption process.** After removing 28 nodes, the original graph has fragmented into cliques of different sizes: one 5-clique, four 3-cliques, and eight 2-cliques, with the remaining nodes being isolated. Since betweenness centrality can no longer discriminate between the nodes, degree centrality will be used to select a node from the 5-clique.

The enhancements discussed, such as the incorporation of secondary ranking criteria and emphasising the LCC, serve to refine our understanding and application of network disruption strategies. These refinements not only optimise existing strategies but also open avenues for customised approaches tailored to specific disruption objectives or network structures. As we transition into the experimental analysis, we will explore how these enhancements impact the effectiveness of each disruption strategy in the targeted criminal network.

The dynamic batch removal disruption process is outlined in Algorithm 1. All code and experiments are available at https://github.com/UTS-CASLab/Disrupt-Criminal-Network.

**Algorithm 1**: Dynamic Batch Removal Disruption

```
input: Graph G, number of nodes N, batch size b, percentage r, bool-
       ean within-LCC
output: R@r
1 initialise: threshold ← N × r, removed ← 0, k ← 0;
2 initialise: LCCSizeAtStep[k] ← getLCCSize(G);
3 for removed < threshold and G.num_of_nodes ≥ b do
4   if within-LCC then
5     LCC ← getLCC(G);
      /* relax the constraint when the LCC size is smaller than b     */
6     if size(LCC) < b then
7       S ← G;
8     else
9       S ← G.subgraph(LCC);
10  else
11    S ← G;
      /* select b nodes via a strategy                                */
12  targetNodes ← select_nodes_to_remove(S, b);
13  G.remove_nodes(targetNodes);
14  removed = removed + len(targetNodes);
15  k = k + b;
16  LCCSizeAtStep[k] ← getLCCSize(G);
```

17 $R@r \leftarrow \frac{sum(LCCSizeAtStep.values()) - LCCSizeAtStep[0]}{N \times (len(LCCSizeAtStep) - 1)}$;

## 5.3 Result and discussion

In the experiment, we implement the seven classic disruption strategies, plus two strategies that utilise a secondary centrality, i.e., DEG-BTW, and BTW-DEG, on the Unified Graph. The disruption is done in a dynamic manner until a percentage of nodes are removed, that is to say, re-ranking nodes after each removal step. Furthermore, we also present the situation where the constraint of disrupting only the LCC is applied. (Notice that the GND algorithm inherently includes the LCC constraint; Additionally, GND cannot calculate $R@40\%$, because the network becomes very sparse in later stages.) The detailed result is shown in Table 3.

First, in the standard setting where target nodes are chosen from the entire graph, APTA and MBA are found to be the most effective strategies across batch sizes, followed by BTW and BTW-DEG. In contrast, CI and CoreHD demonstrate inferior performance in network disruption. Although CI is designed to provide a more comprehensive view of a node's influence by considering its extended neighbourhood and is proposed as an improvement over DEG, it actually fails to identify nodes that are pivotal for maintaining overall network connectivity and falls behind DEG. Take the star graph as an example, CI assigns zeros to all nodes, including the centre of the star, whereas DEG ranks the centre node as the most important one. When it comes to CoreHD, the approach could be effective if the primary goal is to break cycles within the network. However, nodes in cycles may not be the nodes that, if removed,

**Table 3. Dynamic disruption of various batch sizes, measured in *R*@20% and *R*@40%.** Nine disruption strategies are included, performed both on the entire graph and on the LCC (referred to as Standard and W-LCC, respectively). The lowest R-values for each column are put in bold, and the lowest values across two columns are asterisked.

| | Strategy | R@20% | | R@40% | |
|---|---|---|---|---|---|
| | | Standard | W-LCC | Standard | W-LCC |
| b = 1 | DEG | 0.3347 | 0.2773 | 0.1750 | 0.1457 |
| | DEG-BTW | 0.3108 | 0.2773 | 0.1625 | 0.1457 |
| | BTW | 0.2655 | 0.2600 | 0.1440 | 0.1373 |
| | BTW-DEG | 0.2655 | 0.2600 | 0.1418 | 0.1373 |
| | CI | 0.3774 | 0.3324 | 0.1983 | 0.1730 |
| | CoreHD | 0.4323 | 0.3878 | 0.2257 | 0.1999 |
| | APTA | **0.2609** | **0.2566\*** | 0.1439 | **0.1362\*** |
| | MBA | 0.2629 | 0.2612 | **0.1397** | 0.1378 |
| | GND | - | 0.2825 | - | - |
| b = 2 | DEG | 0.3031 | 0.2669 | 0.1578 | 0.1415 |
| | DEG-BTW | 0.2899 | **0.2600\*** | 0.1501 | **0.1376\*** |
| | BTW | 0.2687 | 0.2750 | 0.1434 | 0.1446 |
| | BTW-DEG | 0.2681 | 0.2750 | 0.1412 | 0.1443 |
| | CI | 0.3749 | 0.3341 | 0.1946 | 0.1736 |
| | CoreHD | 0.4242 | 0.3731 | 0.2206 | 0.1926 |
| | APTA | **0.2612** | 0.2606 | 0.1429 | 0.1382 |
| | MBA | 0.2687 | 0.2773 | **0.1407** | 0.1443 |
| | GND | - | 0.2620 | - | - |
| b = 3 | DEG | 0.2695 | **0.2396\*** | 0.1459 | **0.1335\*** |
| | DEG-BTW | 0.2620 | **0.2396\*** | 0.1418 | 0.1339 |
| | BTW | 0.2504 | 0.2637 | 0.1389 | 0.1464 |
| | BTW-DEG | 0.2504 | 0.2637 | 0.1376 | 0.1464 |
| | CI | 0.3458 | 0.3400 | 0.1874 | 0.1849 |
| | CoreHD | 0.3972 | 0.3466 | 0.2168 | 0.1878 |
| | APTA | 0.2529 | 0.2554 | 0.1426 | 0.1418 |
| | MBA | **0.2488** | 0.2637 | **0.1347** | 0.1464 |
| | GND | - | 0.2461 | - | - |
| b = 4 | DEG | 0.2761 | 0.2569 | 0.1482 | 0.1538 |
| | DEG-BTW | 0.2559 | 0.2516 | 0.1386 | 0.1521 |
| | BTW | 0.2217 | 0.2399 | 0.1242 | 0.1349 |
| | BTW-DEG | 0.2217 | 0.2399 | 0.1221 | 0.1349 |
| | CI | 0.3198 | 0.2921 | 0.1738 | 0.1610 |
| | CoreHD | 0.3507 | 0.3166 | 0.1924 | 0.1764 |
| | APTA | 0.2473 | 0.2473 | 0.1370 | 0.1359 |
| | MBA | **0.2175\*** | **0.2388** | **0.1210\*** | **0.1317** |
| | GND | - | 0.2562 | - | - |

would cause the network to fragment into smaller, disconnected components (think again of the example of a star graph).

Next, let us see how the constraint of within-LCC influences the performance of different strategies with different batch sizes. When batch size equals 1, imposing the constraint of within-LCC consistently improves the performance for all disruption strategies, especially for degree-based approaches such as DEG, DEG-BTW, CI and CoreHD. The rationale behind this improvement is straightforward: the primary goal of the disruption process is to minimise the

size of the LCC at each step. Thus, concentrating the disruption efforts on the LCC is intuitively beneficial.

When the batch size increases to two and beyond, the effects of the within-LCC constraint begin to diverge depending on the type of strategy employed. Although an integral statistical analysis on all nine approaches show that there is no significant difference between the standard and within LCC strategies for batch sizes b = 3 and b = 4 (see Table 4), a more detailed analysis can be conducted based on the nature of different disruption approaches.

For degree-based approaches like DEG, DEG-BTW, CI, and CoreHD, the within-LCC constraint consistently enhances performance across various batch sizes (the only exception is at R@40% with a batch size of 4 due to an increased number of connected components of comparable sizes at a later stage). This is likely because degree centrality is inherently a local measure that quantifies a node's immediate influence based on its direct connections. When the scope of the algorithm is narrowed to the LCC, this local effect is magnified. The LCC, being the most connected and structurally significant part of the network, often contains high-degree nodes that are crucial for maintaining the network's largest connected structure. Therefore, focusing on the LCC enables these degree-based methods to target the most impactful nodes more efficiently, further optimising the disruption process.

In contrast, for betweenness-based approaches like BTW, BTW-DEG and MBA, the application of the within-LCC constraint results in worse performance when the batch size is larger than one. The reason behind this is that betweenness centrality relies on global information, and narrowing the focus to the LCC might neglect key bridging nodes that exist outside of it. Overall, our results suggest that when the batch size is relatively small (batch size ranging from 1 to 3), the best disruption strategies are found to be those having within-LCC constraints. Also, degree-based approaches benefit more from a focus on LCC.

Lastly, the result also sheds light on the impact of incorporating a secondary centrality measure. Generally, DEG-BTW outperforms DEG, and BTW-DEG is more effective than BTW. There are instances where the strategies with and without secondary centrality measures yield identical outcomes, especially when a small percentage of nodes are targeted to be removed, as indicated by R@20%. This suggests that in the initial phases, the top-ranked nodes are usually structurally distinct, making the primary centrality measure sufficient for effective node removal. However, as the percentage increases, the benefit of including a secondary centrality becomes more evident. For instance, we observed a decline in the number of identical outcomes between single-metric and dual-metric strategies—from 9 pairs at R@20% to just 4 pairs at R@40%. This indicates that as more nodes are targeted for removal, a dual-metric approach to node ranking becomes increasingly beneficial.

It's noteworthy that degree centrality, due to its computational efficiency, emerges as the most commonly employed secondary ranking criterion. It also plays a pivotal role in CoreHD, as core only identifies a candidate set of nodes, and it is within this set that degree centrality is

**Table 4. Paired t-test results showing the differences between the standard and within LCC approaches for different values of batch size.** A significant p-value ($p \leq 0.05$) indicates a statistically significant difference between the two approaches.

| Batch Size | T-Statistic | P-Value | Degrees of Freedom |
|:---:|:---:|:---:|:---:|
| b = 1 | 4.3800 | < 0.001 | 15 |
| b = 2 | 2.7496 | 0.0149 | 15 |
| b = 3 | 1.1758 | 0.2580 | 15 |
| b = 4 | 0.0609 | 0.9522 | 15 |

employed to finalise the rankings. Moreover, in the absence of articulation points or a core, degree centrality becomes the fallback option for ranking the nodes in APTA and CoreHD.

## 6 Direct optimisation algorithms

The above experiment revealed the effectiveness of introducing a secondary centrality metric as well as including the constraint within the LCC. The best disruption strategies are found to be APTA (when $b = 1$), DEG-based strategies (when $b = 2$ or $3$), and BTW-based strategies (when $b = 4$).

A natural continuation is, of course, to find another disruption strategy that outperforms them. Inspired by the outstanding performance of the greedy approach APTA, and also due to the relatively small size of the graph we manage in our analysis, we propose to first apply a naive greedy disruption strategy on all nodes, not only on the articulation points. Then, based on the findings from the experiment, we propose a novel greedy disruption strategy that also takes advantage of crucial structural features.

### 6.1 A naive greedy disruption approach

In order to conduct a continuous batch removal, at each step, the naive greedy disruption approach (GRD) goes over all possible combinations of all nodes and removes nodes leading to the largest drop in LCC size. The experiment is done on the Unified Graph and the result is reported in Table 5. The improvement of GRD is evident, especially when the batch size is larger than one. When the batch size equals one, the performance of GRD and APTA are quite similar. This can be attributed to the fact that most individual nodes that lead to the largest drop in LCC size are also articulation points. However, as the batch size increases, the advantages of GRD become more pronounced. This suggests that although the greedy approach aims to maximise the disruption effect only at each individual step, it does achieve an overall better performance compared to the best structural-metric-based approaches.

One would expect that the within-LCC constraint should limit the effectiveness of a greedy approach due to a reduced search space. Certainly, there is no difference with $b$ equal to one since a greedy algorithm would naturally target a node within the LCC to achieve a reduction in its size. The difference between with or without the constraint within-LCC is still not significant when $b$ equals 2. As the batch size exceeds two, the performance of the standard GRD significantly outpaces that of the with-in LCC GRD. This implies that, with larger batch sizes, the freedom to remove nodes from multiple components becomes increasingly important for the

**Table 5. Comparison of best-performing strategies at each batch size $b$ with the naive greedy disruption approach (GRD), measured in $R@20\%$ and $R@40\%$.** The lowest R-values are highlighted in bold font.

|  | Strategy | R@20% | | R@40% | |
|---|---|---|---|---|---|
|  |  | **Standard** | **W-LCC** | **Standard** | **W-LCC** |
| b = 1 | APTA | 0.2609 | **0.2566** | 0.1439 | **0.1362** |
|  | GRD | 0.2569 | 0.2569 | 0.1370 | 0.1370 |
| b = 2 | DEG-BTW | 0.2899 | 0.2600 | 0.1501 | 0.1376 |
|  | GRD | **0.2354** | 0.2377 | **0.1247** | 0.1274 |
| b = 3 | DEG-BTW | 0.2620 | 0.2396 | 0.1418 | 0.1339 |
|  | GRD | **0.2056** | 0.2231 | **0.1132** | 0.1269 |
| b = 4 | MBA | 0.2175 | 0.2388 | 0.1210 | 0.1317 |
|  | GRD | **0.1834** | 0.2100 | **0.0997** | 0.1309 |

overall effectiveness of GRD, as opposed to removing all nodes of the batch size within the LCC.

**Algorithm 2**: SF-GRD (at one removal step)

```
input: Graph G, batch size b, top nodes t
output: List of target nodes targetNodes
1 Function SF-GRD(G, b, t)
    /* select top b components to form G_sub                              */
2   G_sub ← topComponents(G, b);
3   min_lcc_size ← getLCCSize(G_sub);
4   targetNodes ← [];
5   search_space ← topNodes_BTW(G_sub, t) ∪ topNodes_DEG(G_sub, t) ∪ top-
    Nodes_AP(G_sub, t);
6   for node_tuple in combinations(search_space, b) do
7     temp_nodes ← G_sub without node_tuple;
8     G_temp ← G.subgraph(temp_nodes);
9     lcc_size_current ← getLCCSize(G_temp);
10     if lcc_size_current <= min_lcc_size then
11       min_lcc_size ← lcc_size_current;
12       targetNodes ← node_tuple;
13   return targetNodes;
```

## 6.2 Structurally filtered greedy disruption

Obviously, GRD's significant improvement in performance comes at the expense of efficiency: the time complexity of looping over the combination of $n$ nodes alone is $O(N^n)$, which is clearly infeasible even for moderate values of $n$. Therefore, we further analyse the rankings and characteristics of three key structural metrics for those nodes found by GRD. To meet the continuous batch removal process, these metrics are recalculated after each removal step. The three metrics, namely, betweenness centrality, degree centrality and articulation points, are selected because they have proven to be the most effective structural-based metrics to disrupt the criminal network. The detailed ranking information is shown in Fig 3. It is clear that a large portion of the nodes found in GRD also have high rankings in betweenness centrality and degree centrality, and the majority of them are identified as articulation points.

This finding inspired us to create a *structurally filtered greedy disruption* strategy (in short, SF-GRD). The SF-GRD algorithm is designed to enhance the efficiency of the naive greedy disruption approach by constraining the search space to nodes that exhibit specific structural characteristics. Specifically, SF-GRD targets a union of nodes that either rank highly in betweenness centrality, excel in degree centrality, or are identified as articulation points. By focusing only on these high-ranking nodes, SF-GRD reduces the combinatorial search space, significantly lowering the computational complexity. Moreover, to maintain a constant-size search space, we limit our selection to the top-ranking articulation points, sorted by their degree centrality. This targeted approach allows SF-GRD to efficiently obtain approximate solutions that are close in effectiveness to the optimal solutions found by the naive greedy method. The SF-GRD algorithm is formally described in Algorithm 2. It determines the target nodes to be removed in a single disruption step. For a continuous batch removal process, the function select_nodes_to_remove in Algorithm 1 can be replaced by the SF-GRD function.

The performance outcomes for SF-GRD, GRD, and the top structure-metric-based methods are summarised in Table 6. Overall, SF-GRD excels over structure-metric-based strategies and achieves comparable effectiveness to that of GRD. Like GRD, SF-GRD also benefits from relaxing the within-LCC constraint when the batch size is larger than one. In order to gain a deeper understanding of the disruption process, we take the case of *R*@20% and *b* = 3 as an example and plot how each step of removal affects the size of the LCC, when different

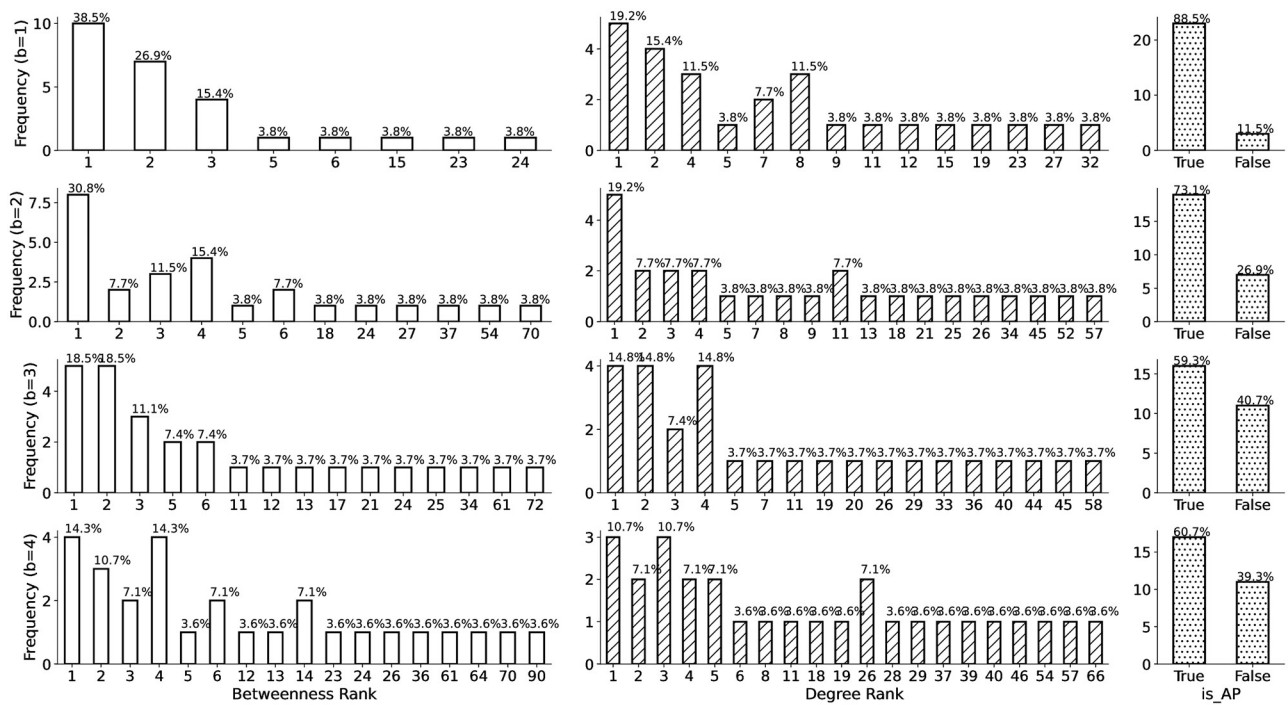

**Fig 3. Structural characteristics of the nodes removed by GRD.** A large percentage of nodes, across different batch sizes, rank highly in the betweenness centrality and degree centrality, and are also articulation points.

disruption strategies are applied (Fig 4). Under this setting, SF-GRD performs closely to GRD, and both of them outperform DEG-BTW by a large margin. After the removal of 27 nodes, all three approaches have reached a similar disruption effect, with the size of LCC dropping to around 5 and 6. That also explains why we should focus more on the initial disruption steps.

Furthermore, we examine the roles of the removed nodes identified by those approaches (Table 7). We focus on the first three removal steps with a batch size equal to 3 and report the

**Table 6. Performance comparison of the best performing structure-metric-based approach, GRD, and SF-GRD, measured in *R*@20% and *R*@40%.** The number of highest-ranking nodes *t* is set to be 5 in the experiment.

|  | Strategy | *R*@20% | | *R*@40% | |
|---|---|---|---|---|---|
|  |  | Standard | W-LCC | Standard | W-LCC |
| b = 1 | APTA | 0.2609 | 0.2566 | 0.1439 | 0.1362 |
|  | GRD | 0.2569 | 0.2569 | 0.1370 | 0.1370 |
|  | SF-GRD | 0.2569 | 0.2569 | 0.1370 | 0.1370 |
| b = 2 | DEG-BTW | 0.2899 | 0.2600 | 0.1501 | 0.1376 |
|  | GRD | 0.2354 | 0.2377 | 0.1247 | 0.1274 |
|  | SF-GRD | 0.2382 | 0.2474 | 0.1249 | 0.1324 |
| b = 3 | DEG-BTW | 0.2620 | 0.2396 | 0.1418 | 0.1339 |
|  | GRD | 0.2056 | 0.2231 | 0.1132 | 0.1269 |
|  | SF-GRD | 0.2065 | 0.2247 | 0.1119 | 0.1269 |
| b = 4 | MBA | 0.2175 | 0.2388 | 0.1210 | 0.1317 |
|  | GRD | 0.1823 | 0.2100 | 0.1034 | 0.1309 |
|  | SF-GRD | 0.2111 | 0.2409 | 0.1146 | 0.1322 |

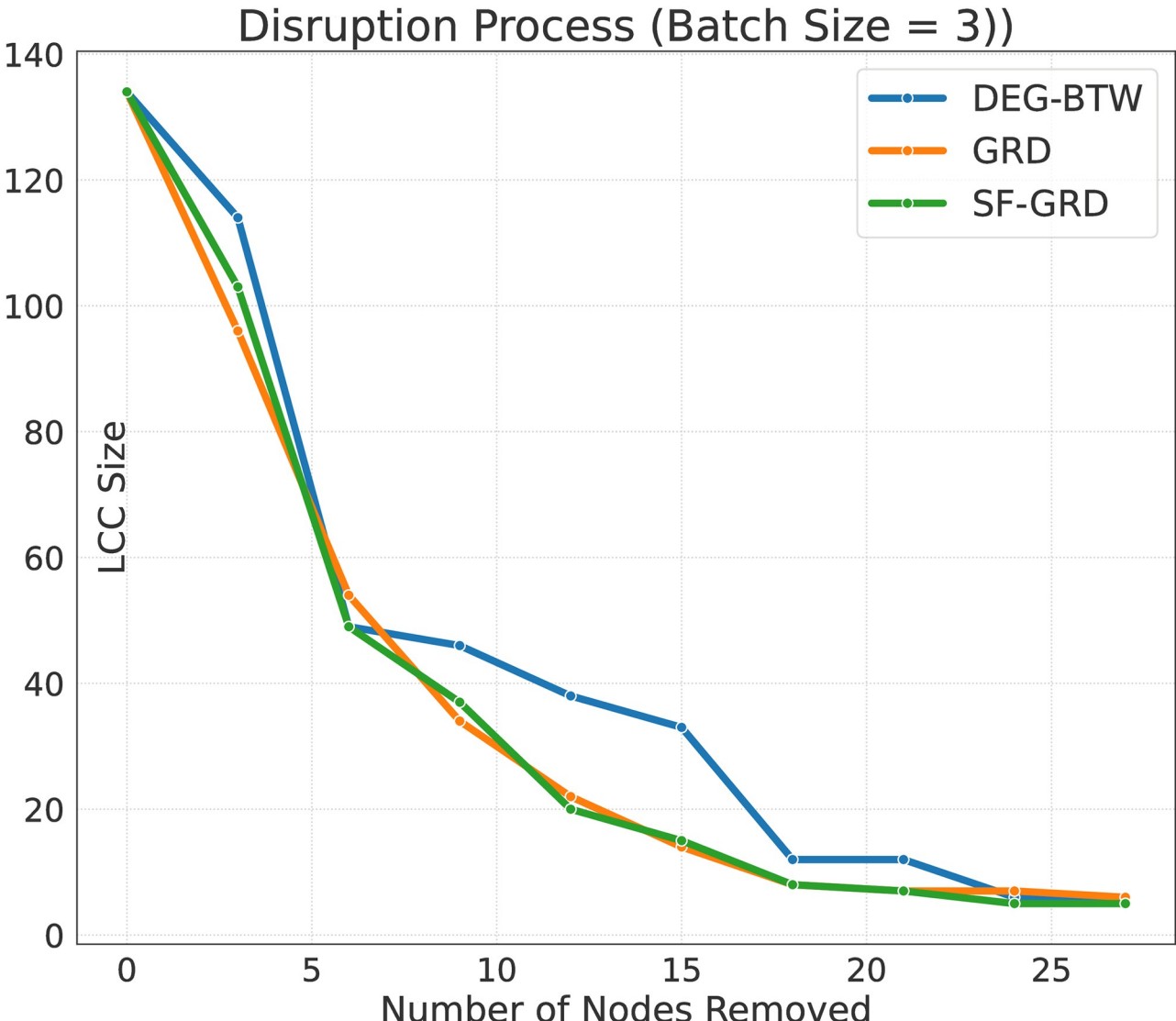

**Fig 4. A detailed disruption process.** With the batch size setting to 3, 27 nodes are removed in 9 steps.

**Table 7. Detailed node categories using different disruption approaches, highlighting the first three steps for b = 3.** Nodes marked with 'F' belong to mafia families, while 'L' denotes a leadership role.

|  | DEG-BTW | GRD | SF-GRD |
|---|---|---|---|
| Step 1 | N18 (F&L), N47 (F&L), N68 (F&L) | N18 (F&L), N68 (F&L), N25 (F&L) | N61 (F&L), N68 (F&L), N18 (F&L) |
| Step 2 | N61 (F&L), N27 (F&L), N22 | N22, N27 (F&L), N61 (F&L) | N22, N47 (F&L), N27 (F&L) |
| Step 3 | N12 (F), N11 (L), N29 | N43, N29, N69 | N48 (F), N45 (F), N51 (F) |

nodes removed together with their roles—whether they are from the two mafia clans or they take a leadership role. We can see that nodes associated with both mafia families and leadership roles (marked with *F&L*) are universally targeted by all three disruption strategies within the initial two steps. In fact, 5 out of the 6 nodes removed in these early stages possess such dual roles. Nodes that are important to hold the network together, also hold important roles in the organisation. Our finding aligns with the intuition that we should first take out the leaders in order to disrupt the network. Certainly, the importance of certain nodes might diminish along with the decomposition of the network, which explains why GRD targets the nodes without specific roles at the third step. In a disrupted network, nodes previously less influential may take on new significance, while the influence of originally key nodes may wane.

## 6.3 Time complexity analysis

Lastly, we discuss the time complexity of the proposed SF-GRD approach and compare it with GRD and BTW. At each step, SF-GRD involves first the calculation and ranking of three structural metrics, which are the betweenness centrality, the degree centrality, and the degree ranked articulation points. Among them, the dominating computational cost comes from the betweenness centrality, which is $O(VE + V^2 \log V)$ using the Brandes algorithm [39]. Then, in the greedy search loop, the total number of combinations is $\binom{t'}{b}$, and the calculation of the size of the LCC is $(V + E)$. Here, $t'$ is the size of the search space, equal to 3 times $t$, where $t$ is the number of nodes selected for each of the three structural metrics (see Algorithm 2). Therefore, the time complexity for SF-GRD is $O(VE + V^2 \log V + \binom{t'}{b} \cdot (V + E))$. Since $t'$ and $b$ are usually set to be small integers (in our experiment, $t'$ is set to be 15, and $b$ is capped at 4), the time complexity for large graph would degrade to that of the betweenness centrality, i.e., $O(VE + V^2 \log V)$.

In comparison, GRD is looping over all combinations of node tuples from the entire graph, leading to a prohibitive time complexity of $O(\binom{V}{b} \cdot (V + E))$. The comparison of the time complexity and the actual running time is given in Table 8. We used a machine with an Intel Xeon 6238R CPU at a base frequency of 2.2 Hz to run the experiment. We observe that SF-GRD is substantially more efficient than GRD while maintaining only a slightly higher computational cost compared to BTW. Specifically, with $b = 4$, SF-GRD is more than four orders of magnitude faster than GRD.

In short, through a structurally-filtered scope, SF-GRD has brought down the combinatorial complexity of GRD to a polynomial time complexity while still achieving a comparable performance. SF-GRD successfully combines the best of both worlds: the effectiveness of GRD-like methods and the efficiency of structural-metric methods.

**Table 8. Time complexity and actual experimental time comparison.** $N$ is the number of nodes in a graph, $E$ is the number of edges, $b$ is the batch size, and $t'$ is the size of the search space, set to be 15 in the experiment.

| Method | Time Complexity (One Step) | Actual Time (s) | |
| --- | --- | --- | --- |
| | | $b = 3$ | $b = 4$ |
| BTW | $O(NE + N^2 \log N)$ | 0.129 | 0.102 |
| GRD | $O(\binom{N}{b} \cdot (N + E)))$ | 475 | 12567 |
| SF-GRD | $O(NE + N^2 \log N + \binom{t'}{b} \cdot (N + E))$ | 0.393 | 0.467 |

## 7 Conclusion

We first introduced a generalised robustness metric suitable for partial and continuous batch removal processes. We then evaluated several disruption strategies on a Sicilian Mafia network and proposed a within-LCC constraint which mimics the police raid operations. We revealed that this constraint would limit the effectiveness of the betweenness-based approaches but would enhance the performance of degree-based ones with a moderate batch size. Furthermore, we proposed a structurally-filtered greedy disruption strategy based on the structural characteristics of the nodes removed by a naive greedy approach. We found that the new algorithm significantly outperformed all classic disruption strategies while maintaining the same time complexity as the betweenness centrality approach.

In addition to applying our proposed approaches to different types of networks, a particularly interesting direction for future research involves the integration of more sophisticated features that capture higher-order interactions within the network. For example, leveraging motif-based centrality measures could provide deeper insights into the roles that specific patterns of interconnections play in maintaining the network's cohesion [40, 41]. Another research avenue is to incorporate the metrics that shed light on the formation of fundamental structures such as triangles and quadrangles [42–44], and investigate how networks with high versus low clustering/quadrangle coefficients respond to different disruption strategies.

## Author Contributions

**Conceptualization:** Mingshan Jia, Katarzyna Musial.

**Data curation:** Pasquale De Meo, Katarzyna Musial.

**Formal analysis:** Mingshan Jia.

**Funding acquisition:** Bogdan Gabrys, Katarzyna Musial.

**Investigation:** Mingshan Jia, Pasquale De Meo.

**Methodology:** Mingshan Jia, Pasquale De Meo, Katarzyna Musial.

**Project administration:** Bogdan Gabrys, Katarzyna Musial.

**Resources:** Bogdan Gabrys, Katarzyna Musial.

**Software:** Mingshan Jia.

**Supervision:** Pasquale De Meo, Bogdan Gabrys, Katarzyna Musial.

**Validation:** Mingshan Jia.

**Visualization:** Mingshan Jia.

**Writing – original draft:** Mingshan Jia.

**Writing – review & editing:** Mingshan Jia, Pasquale De Meo, Bogdan Gabrys, Katarzyna Musial.

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
