## [Decision Letter · Decision Letter 0]

21 Feb 2024

PONE-D-23-41294Network disruption via continuous batch removal: The case of Sicilian MafiaPLOS ONE

Dear Dr. Jia,

Thank you for submitting your manuscript to PLOS ONE. After careful consideration, we feel that it has merit but does not fully meet PLOS ONE’s publication criteria as it currently stands. Therefore, we invite you to submit a revised version of the manuscript that addresses the points raised during the review process.

We look forward to receiving your revised manuscript.

Kind regards,

Giacomo Fiumara, PhD

Academic Editor

PLOS ONE

Journal Requirements:

"This work was supported by the Australian Research Council, Grant No. DP190101087: Dynamics and Control of Complex Social Networks."

"This work was supported by the Australian Research Council, Grant No. DP190101087: “Dynamics and Control of Complex Social Networks”. "

"This work was supported by the Australian Research Council, Grant No. DP190101087: Dynamics and Control of Complex Social Networks."

4. Thank you for uploading your study's underlying data set. Unfortunately, the repository you have noted in your Data Availability statement does not qualify as an acceptable data repository according to PLOS's standards.

Reviewers' comments:

Reviewer's Responses to Questions

**Comments to the Author**

1. Is the manuscript technically sound, and do the data support the conclusions?

Reviewer #1: No

Reviewer #2: Partly

2. Has the statistical analysis been performed appropriately and rigorously? 

Reviewer #1: N/A

Reviewer #2: No

3. Have the authors made all data underlying the findings in their manuscript fully available?

Reviewer #1: Yes

Reviewer #2: Yes

4. Is the manuscript presented in an intelligible fashion and written in standard English?

Reviewer #1: Yes

Reviewer #2: No

5. Review Comments to the Author

Reviewer #1: The research article titled "Network disruption via continuous batch removal: The case of Sicilian Mafia" has no novelty in suggested research work. The issue is already resolved by researchers. I have recommended to work on different dataset instead of using a single dataset.

Reviewer #2: Overall, the study presents disruption strategies and evaluates their effectiveness on a real-world network, offering insights into how network structure influences disruption outcomes. Additionally, it highlights the potential for future research to delve deeper into understanding network dynamics and improving disruption strategies further. However, following things must be added in the revised paper.

The study begins by introducing a novel generalized robustness metric tailored for partial and continuous batch removal processes. It is noteworthy that all introduced nomenclature should be clearly defined either in the introduction or upon their first mention to facilitate understanding for readers unfamiliar with the terminology.

Subsequently, the authors evaluated several disruption strategies on a Sicilian Mafia network. While they claimed the robustness of their approach, there is a lack of comparative analysis with existing research to substantiate this assertion. To address this gap, a thorough comparison with other relevant works is warranted. This would involve examining how the proposed strategies fare against those from prior studies in terms of effectiveness, efficiency, and robustness under similar conditions.

Furthermore, the study proposed a within-LCC constraint simulating police raid operations. This constraint was found to impact the effectiveness of disruption strategies, favoring degree-based approaches with moderate batch sizes over betweenness-based ones. However, to provide a more comprehensive evaluation, additional statistical analysis of the results is necessary. This analysis should include measures such as confidence intervals, significance tests, and potentially sensitivity analyses to assess the robustness of the findings under different conditions.

Moreover, the authors introduced a structurally-filtered greedy disruption strategy based on the characteristics of nodes removed by a naive greedy approach. This strategy reportedly outperformed classic disruption strategies while maintaining similar time complexity to betweenness centrality. To strengthen the validity of this claim, it is essential to conduct a more rigorous comparison with existing approaches. This could involve assessing the performance of the proposed strategy against a wider range of benchmarks and providing a detailed discussion of the results.

In addition to addressing these points, future research directions could involve exploring how other structural characteristics of networks can offer valuable insights into network disruption. This could lead to a deeper understanding of network dynamics and the development of even more effective disruption strategies.

6. PLOS authors have the option to publish the peer review history of their article (what does this mean?). If published, this will include your full peer review and any attached files.

Reviewer #1: No

Reviewer #2: **Yes: **Ankit shah

---

## [Author Response · Author response to Decision Letter 0]

26 Mar 2024

Please find detailed reponse in the attached rebuttal letter.

---

## [Decision Letter · Decision Letter 1]

22 Jul 2024

PONE-D-23-41294R1Network disruption via continuous batch removal: The case of Sicilian MafiaPLOS ONE

Dear Dr. Jia,

Thank you for submitting your manuscript to PLOS ONE. After careful consideration, we feel that it has merit but does not fully meet PLOS ONE’s publication criteria as it currently stands. Therefore, we invite you to submit a revised version of the manuscript that addresses the points raised during the review process.

We look forward to receiving your revised manuscript.

Kind regards,

Giacomo Fiumara, PhD

Academic Editor

PLOS ONE

Journal Requirements:

Additional Editor Comments:

Dear authors,

The second round of the reviews of the manuscript "Network disruption via continuous batch removal: The case of Sicilian Mafia" has finally ended. My decision is that the manuscript has to undergo a minor revision before considering it for publication in PLOS ONE. The contribution of Reviewer 1 clearly refers to another manuscript. We tried to contact him and ask to change it, but with no success. The review of Reviewer 3 does not fully target the main goal of the paper and his suggestions must be considered only in part. In particular, I suggest to address the following comments:

1. Insufficient explanation of robustness

4. Dealing with NP-hardness

5. Relationship between noise removal and feature reduction

6. Validity of evaluation metrics

while the remaining comments could be useful for future works on this interesting theme. I also suggest to exclude the challenges proposed by Reviewer 3 as they do not exactly pertain to the research illustrated in the paper.

Reviewers' comments:

Reviewer's Responses to Questions

**Comments to the Author**

1. If the authors have adequately addressed your comments raised in a previous round of review and you feel that this manuscript is now acceptable for publication, you may indicate that here to bypass the “Comments to the Author” section, enter your conflict of interest statement in the “Confidential to Editor” section, and submit your "Accept" recommendation.

Reviewer #1: (No Response)

Reviewer #2: All comments have been addressed

Reviewer #3: All comments have been addressed

Reviewer #4: All comments have been addressed

2. Is the manuscript technically sound, and do the data support the conclusions?

Reviewer #1: Yes

Reviewer #2: Yes

Reviewer #3: Partly

Reviewer #4: Yes

3. Has the statistical analysis been performed appropriately and rigorously? 

Reviewer #1: Yes

Reviewer #2: Yes

Reviewer #3: No

Reviewer #4: Yes

4. Have the authors made all data underlying the findings in their manuscript fully available?

Reviewer #1: Yes

Reviewer #2: Yes

Reviewer #3: Yes

Reviewer #4: Yes

5. Is the manuscript presented in an intelligible fashion and written in standard English?

Reviewer #1: (No Response)

Reviewer #2: Yes

Reviewer #3: Yes

Reviewer #4: Yes

6. Review Comments to the Author

Reviewer #1: Based on the abstract and key sections of the manuscript titled "G-Net: Implementing an Enhanced Brain Tumor Segmentation Framework using Semantic Segmentation Design," here are my review comments

1. Include comparisons with architectures that do not incorporate features like Self-Attention and Squeeze Excitation blocks to highlight G-Net’s unique contributions to brain tumor segmentation.

2. Quantify the model's precision improvement in tumor boundary localization and detail capture through metrics or visual examples, providing a clearer benchmark of novelty.

3. Add statistical analyses of the dataset’s diversity, including age distribution, tumor sizes, and imaging modalities, to demonstrate the model's robustness across varied cases.

4. .Cite below relevant articles in related work to improve . DOI: 10.1109/ICTC58733.2023.10392830 DOI: 10.1109/ACCESS.2023.3330919 https://doi.org/10.3390/math11194189
https://doi.org/10.3390/diagnostics13162650
https://itiis.org/digital-library/90390

5. Conduct a side-by-side performance evaluation with state-of-the-art methods using standard datasets, focusing on accuracy, sensitivity, and specificity.

6. Enhance clarity by providing flow diagrams or pseudocode, offering readers a visual understanding of the feature extraction process and block integration.

7. Detail a pilot study or case studies where G-Net has been integrated into diagnostic workflows, including any operational challenges and solutions.

8. Outline a specific roadmap for scalability, including preliminary experiments on other tumor types or explorations into transfer learning for adapting G-Net to other medical imaging tasks.

9. Include a section discussing alternative configurations tested during development, explaining why certain designs were favored over others based on performance metrics.

10. Discuss the impact of open-source contributions on the research community, mentioning any collaborative projects or external validations that have leveraged the work.

Reviewer #2: All the queries are addressed by the author. Comparative analysis is included in the revised manuscript.

Reviewer #3: Comments:

This Paper and Datasets are very interesting paper. However, I believe it could be further refined by not only focusing on the current method but also by incorporating new approaches and revisions. These would highlight important aspects of criminal network analysis.

1. Insufficient explanation of robustness:

The concept of robustness in network analysis methods is not sufficiently explained in this paper. In particular, it is necessary to clearly demonstrate, with theoretical justification, how the proposed Greedy Disruption Approach (GRD) improves robustness.

2. Lack of mathematical formalization:

The explanation of the proposed methods, including GRD, relies mainly on program code and lacks mathematical formalization. It is necessary to clearly show the theoretical basis of the algorithm using equations and analyze its properties (convergence, computational complexity, etc.).

3. Insufficient consideration of practicality:

There is insufficient discussion about the effectiveness of the proposed method against more complex criminal activities such as countering intelligent criminals, impersonation, and behaviors that induce attribution of crimes to general clusters. The strengths and limitations of how the proposed method can be applied to these real scenarios should be clearly shown.

4. Dealing with NP-hardness:

While it is stated that the network disruption task is NP-hard, the explanation of how this difficulty is addressed is insufficient. In particular, it is necessary to explain in detail how the proposed method mitigates this computational difficulty or how it efficiently obtains approximate solutions.

5. Relationship between noise removal and feature reduction:

There is a possibility that removing nodes from the network may simultaneously reduce potentially important features. Consideration of this point is lacking. The theoretical basis and experimental verification of how to balance noise removal and retention of important structural information should be shown.

6. Validity of evaluation metrics:

While the reduction in LCC size is used as the main evaluation metric, there is insufficient discussion about whether this metric appropriately reflects the vulnerability of actual criminal networks. It is necessary to explain in more detail the comparison with other possible evaluation metrics and the reason for choosing this metric.

7. Handling complex network structures:

Actual criminal networks are likely to have more complex structures, but there is insufficient explanation of how the proposed method deals with such complexity. In particular, the applicability to dynamically changing networks and multi-layer networks should be discussed.

Unless these points are appropriately improved, serious questions remain about both the theoretical foundation and practical value of the paper. In particular, it is strongly recommended to improve mathematical rigor, verify effectiveness against practical scenarios, and clarify the positioning of the proposed method in the context of broader criminal network analysis.

Within the scope of this paper, we propose the following analytical methods to address the challenge of detecting intelligent criminals who may mimic ordinary individuals or use patterns that are difficult to detect:

1. Temporal Community Structure Analysis:

a) Theoretical framework:

- Apply dynamic community detection algorithms to analyze evolving community structures over time.

- Identify and analyze differences between stable and temporary communities.

b) Methodology:

- Implement a temporal extension of the Louvain algorithm used in the paper.

- Quantify community persistence and variability.

c) Application examples:

- Analyze temporal change patterns in communities within phone communication and meeting graphs.

- Identify unnaturally stable or unstable communities that deviate from typical community patterns.

2. Multi-layer Network Analysis:

a) Theoretical framework:

- Treat phone communication and meeting graphs as separate layers and analyze inter-layer interactions.

- Detect discrepancies or abnormal matching patterns between layers.

b) Methodology:

- Apply multiplex network models.

- Conduct correlation analysis of centrality measures across layers.

- Perform cross-layer community detection.

c) Application examples:

- Detect unnatural matches or mismatches between phone communication and meeting patterns.

- Identify nodes with divergent roles across layers.

3. Similarity-based Anomaly Detection:

a) Theoretical framework:

- Define similarity metrics for nodes and subgraphs to detect overly similar patterns or unnaturally different patterns.

- Quantify structural similarities between subgraphs using graph kernel methods.

b) Methodology:

- Analyze subgraph similarity using graph edit distance or graph kernels.

- Analyze role similarity of nodes (extension of structural equivalence).

- Detect nodes or subgraphs with abnormally high similarity.

c) Application examples:

- Identify criminal patterns that are excessively similar to general communication patterns.

- Detect subnetworks with unnaturally uniform structures.

4. Entropy-based Anomaly Detection:

a) Theoretical framework:

- Calculate network structure entropy to identify areas with unnaturally low or high entropy.

- Compare local and global entropy.

b) Methodology:

- Compute graph entropy (degree distribution entropy, random walk entropy, etc.).

- Analyze temporal changes in local entropy.

c) Application examples:

- Detect excessively regular (low entropy) or disordered (high entropy) communication patterns.

- Identify subnetworks with unnatural entropy changes over time.

Regarding dataset splitting and validation methods:

The paper appears to treat the entire dataset as a single set for evaluating the proposed method. However, it is essential to split the dataset into training and test sets to confirm the generalizability of the method and avoid overfitting. This division should be made, and performance evaluation should be conducted on the test set. Also, it is strongly recommended to apply methods such as k-fold cross-validation to maximize the use of limited datasets. This can increase the reliability and reproducibility of the results.

It is recommended to compare the proposed method with existing methods using evaluation metrics other than LCC size reduction (e.g., network efficiency, changes in centrality metrics). This can more clearly demonstrate the relative advantage of the proposed method. It is also necessary to verify and report the statistical significance of the obtained results using appropriate statistical tests. Furthermore, it is strongly recommended to conduct a more detailed sensitivity analysis on the main parameters of the proposed method, especially the number of nodes to be removed. Specifically:

a) Provide graphs or tables quantitatively showing the relationship between the number of nodes removed and detection accuracy.

b) Propose a methodology for determining the optimal number of nodes to remove and explain its theoretical basis.

c) Analyze the differences in detection performance between cases with few and many nodes removed, and consider the reasons for these differences.

Please provide a detailed analysis of how effectively the proposed method can identify more sophisticated, difficult-to-detect anomaly patterns. Consider the following points:

a) Evaluate the ability to detect abnormal patterns that cleverly mimic general network structures.

b) Analyze the detection capability for temporally changing anomaly patterns or distributed anomaly patterns spanning multiple subnetworks.

c) Clearly distinguish the types of anomaly patterns that the proposed method can detect particularly effectively and those that are difficult to detect, and explain the reasons for this.

Evaluate the robustness of the proposed method against extreme network structures and special cases:

a) Analyze performance in very dense or very sparse networks.

b) Investigate behavior in networks with extremely high (or low) scale-free or small-world properties.

c) Compare performance in these edge cases with performance in typical cases, and clarify the limits of application of the method.

Reviewer #4: The authors made a carefully revision of their manuscript according to the Reviewers' comments. Therefore, I recommend it for publication in its current form.

7. PLOS authors have the option to publish the peer review history of their article (what does this mean?). If published, this will include your full peer review and any attached files.

Reviewer #1: No

Reviewer #2: No

Reviewer #3: **Yes: **Yasuko Kawahata

Reviewer #4: No

---

## [Author Response · Author response to Decision Letter 1]

22 Jul 2024

Reponses are provided in the attached rebuttal letter.

---

## [Editor Report · Decision Letter 2]

30 Jul 2024

Network disruption via continuous batch removal: The case of Sicilian Mafia

PONE-D-23-41294R2

Dear Dr. Mingshan Jia,

We’re pleased to inform you that your manuscript has been judged scientifically suitable for publication and will be formally accepted for publication once it meets all outstanding technical requirements.

Kind regards,

Giacomo Fiumara, PhD

Academic Editor

PLOS ONE

Additional Editor Comments (optional):

The manuscript has been significantly improved in the various revisions. All concerns raised by the Reviewers have been fully addressed. The manuscript can be accepted for publication in PLOS ONE.
---

## [Editor Report · Acceptance letter]

1 Aug 2024

PONE-D-23-41294R2 

PLOS ONE

Dear Dr. Jia, 

I'm pleased to inform you that your manuscript has been deemed suitable for publication in PLOS ONE. Congratulations! Your manuscript is now being handed over to our production team.

Kind regards, 

on behalf of

Dr. Giacomo Fiumara 

Academic Editor

PLOS ONE